# CaMP: Causal Multi-policy Planning for Interactive Navigation in Multi-room Scenes

**Xiaohan Wang**
Institute of Artificial Intelligence and Robotics
Xi'an Jiaotong University
wuhanwxh2016@stu.xjtu.edu.cn

**Yuehu Liu**
Institute of Artificial Intelligence and Robotics
Xi'an Jiaotong University
liuyh@mail.xjtu.edu.cn

**Xinhang Song**
Institute of Computing Technology
Chinese Academy of Sciences
University of Chinese Academy of Science
xinhang.song@vipl.ict.ac.cn

**Beibei Wang**
Institute of Artificial Intelligence and Robotics
Xi'an Jiaotong University
wangbb@stu.xjtu.edu.cn

**Shuqiang Jiang**
Institute of Computing Technology
Chinese Academy of Sciences
University of Chinese Academy of Science
sqjiang@ict.ac.cn

## Abstract

Visual navigation has been widely studied under the assumption that there may be several clear routes to reach the goal. However, in more practical scenarios such as a house with several messy rooms, there may not. Interactive Navigation (InterNav) considers agents navigating to their goals more effectively with object interactions, posing new challenges of learning interaction dynamics and extra action space. Previous works learn single vision-to-action policy with the guidance of designed representations. However, the causality between actions and outcomes is prone to be confounded when the attributes of obstacles are diverse and hard to measure. Learning policy for long-term action planning in complex scenes also leads to extensive inefficient exploration. In this paper, we introduce a causal diagram of InterNav clarifying the confounding bias caused by obstacles. To address the problem, we propose a multi-policy model that enables the exploration of counterfactual interactions as well as reduces unnecessary exploration. We develop a large-scale dataset containing 600k task episodes in 12k multi-room scenes based on the ProcTHOR simulator and showcase the effectiveness of our method with the evaluations on our dataset.

## 1 Introduction

Embodied AI, incorporating internet intelligence such as vision, language, and reasoning into an embodiment, aims to deploy autonomous agents in real-world environments [16]. In Visual Navigation, a cornerstone task, agents navigate to the goal in static environments where the movement of scene objects is avoided. In a practical scenario an indoor robot may need to navigate in a house with multiple rooms and get blocked by obstacles(see Figure 1). Interactive Navigation (InterNav)[39, 36] considers navigating more efficiently with object interaction rather than merely moving forward and turning around.

37th Conference on Neural Information Processing Systems (NeurIPS 2023).

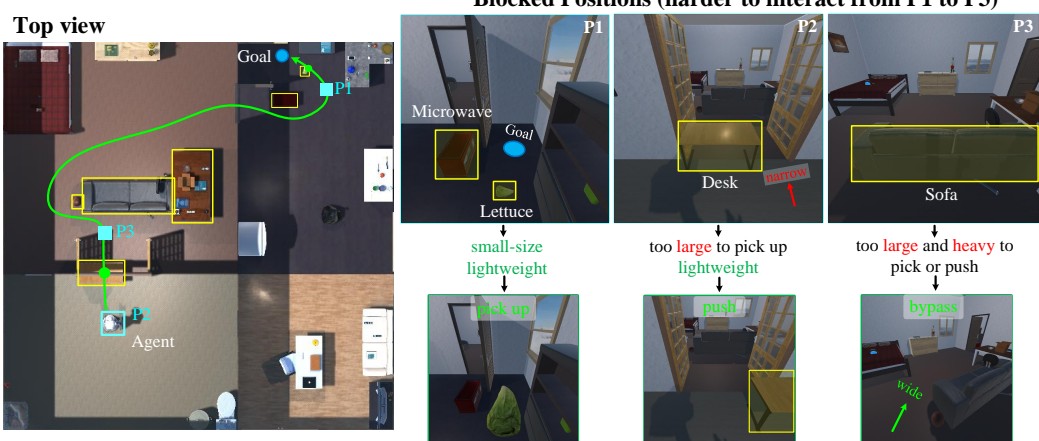

Figure 1: Illustration of InterNav in a 4-room scene. Trying to reach the goal as a blue circle, the agent is blocked by obstacles in yellow boxes spawned in three rooms. The agent needs to make decisions at three blocked positions marked in blue boxes: to bypass or to interact with obstacles? The outcomes depend on the interactive attributes of obstacles (whether they can be pushed away or picked up, e.g. heavy, small, etc.). The green arrow is a path the agent may take to reach the target and the nodes denote interaction with obstacles. Markings in colors are invisible to the agent.

Years of visual navigation exploration witnessed the growing interest in long-horizon navigation in complex environments [13]. Zeng et al.[39] build a dataset benchmarking interactive navigation task on the AI2-THOR simulator, where agents receive visual observation and output the sequence of discrete actions. New challenges are raised compared to visual navigation: First, the agent needs to understand the outcome of object interaction. Second, the need for interaction with obstacles leads to a larger action space for reinforcement learning (RL). However, the dataset is limited to a simple scenario where a few objects are lined up between the starting and target point in a single room. Navigation in multi-room scenes requires agents to plan longer action sequences and understand more diverse states of the environment.

Since the dynamics of object interaction follow the physical rules that are structurally invariant in the environment, there naturally exists a cause-effect relation between the action and its outcome. From the perspective of causality, an RL agent performs interventions to the environment (by taking actions) and estimates the effect (rewards) in order to learn a policy that maximizes the long-term reward. However, learning causality through RL training is challenging due to the existence of unobserved confounders (UCs), which affect both the decision-making of actions and the generation of rewards [41, 18]. In InterNav, the agent may decide to navigate or interact based on whether it is blocked by an object (referred to as an obstacle) whose distribution is unobserved, namely unmeasurable. The existence or type of obstacles is also decisive to the outcome (see Figure 1). For instance, when blocked by a desk the agent may choose to push it aside since empirically it leads to a high reward, while the interaction will turn out otherwise if the desk gets stuck. Without knowing the true causality in interaction, the agent is prone to learn a sub-optimal policy. Prior work [39] on InterNav attempts to predict the change of object pose as a representation to understand the interaction outcomes. The model performance relies on effective obstacle detection and being familiar with obstacle distribution, which may not work in complex scenes with various obstacles. Long-term planning in complex environments also leads to inefficient exploration. Moving forward at blocked positions or trying to interact with no object around makes no difference to the environment and the agent can learn little about the causality from those experiences. Common solutions to visual navigation [35] and InterNav [39] train single policy with end-to-end RL, which requires extensive training to learn a universal policy tackling various scenarios. However, an agent may come up with multiple intents but can only conduct one behavior at once due to its embodiment. In a short-term view, the agent can either interact with objects or navigate somewhere. Hence it is more efficient to learn separate policies for interaction and navigation.

In this paper, we introduce a structural causal model, analyzing InterNav through causal lenses and clarifying the key challenge as the confounding bias caused by obstacles. We show that learning a policy conditioned on its primary decision (referred to as "intent") can enable the agent to learn from counterfactual experiences and make better decisions. As a solution to InterNav, we propose a Causal Multi-Policy model (CaMP), which reduces unnecessary exploration by breaking down the task into simple sub-tasks and enables counterfactual exploration by trialing conditioned on the agent's intent. In particular, three action policies (i.e. navigation, pushing interaction, and picking interaction) are developed by training three actor-critic networks in designed auxiliary tasks. We then train a global policy that receives both observation and action intent and calls action policies in sequence. An intent policy is developed to duplicate the decision of the global policy so that action policies can be called to produce their action intents. To study the InterNav in more practical scenarios, we collect a dataset containing 600k task episodes with various obstacle arrangements, based on the ProcTHOR simulator [14] of large-scale multi-room scenes. We also introduce a new metric measuring the time efficiency of task completion due to the need for interaction in InterNav. Evaluations on our dataset showcase the effectiveness of the proposed model CaMP.

## 2 Related Works

**Interactive Navigation** comes up as the downstream task of visual navigation, which has been extensively studied in Embodied AI literature in the past decade. Most works address PointNav [5, 27, 35] and ObjNav [4, 6, 44, 42, 43], while others develop Visual-language Navigation [1, 29, 33, 21], AudioNav [8, 7], etc that bring in new intelligence components. Environments of sim-to-real [13] or large-scale [14] indoor scenes are also proposed for more complexity. Zeng et al. [39] focus on vision-based agents interacting with directional pushes on iTHOR environment [20]. However, they consider elementary scenarios where obstacles are spawned in a single room for agents to push out of the way. In this work we construct a new dataset based on ProcTHOR environment [14] that provides 12k multi-room scenes with diverse floorplans and various assets, supporting large-scale training.

**Causal Reinforcement Learning.** Causal inference [24, 25, 26] has been a longstanding methodology adopted to pursue the causal effect. Recent years have witnessed the progress of applying causal theory in computer vision tasks including visual question answering [23, 37, 9], scene graph generation [32, 10], visual recognition [22, 31, 37, 38], etc. Causal reinforcement learning (CRL) [18, 17, 40, 3, 41] is motivated by the challenges of RL including data inefficiency and lack of interpretability, and has been applied in traditional RL tasks such as multi-armed bandit problem [3], Markov decision process [41]. Naturally, the causal bias also exists in Embodied AI tasks, but there exists little work that helps train RL agents to complete tasks applying the theory of causal reasoning. We make the first attempt to model the causalities among the interactive navigation and address the confounding bias of obstacles by implementing a counterfactual policy.

**Hierarchical Models for Embodied AI.** To address complex and long-horizon tasks, one of the most popular solutions is the hierarchical approach inspired by Hierarchical Reinforcement learning (HRL) [30, 15]. The main idea is to divide the whole task into solvable sub-tasks through temporal abstraction which allows taking actions at different time scales [2]. Previous attempts applied on embodied question answering [11, 12], and interactive question answering [19] mainly follow the flavor of finding sub-goals for the low-level module to achieve. However, learning policy associated with sub-goals can be data expensive and requires much handcraft design for vision-based tasks. Our model is different from those works following the idea of the option framework [2] that develops multiple policies for different task situations without the need for any sub-goals.

## 3 A Causal View of InterNav

We start by grounding the problem of InterNav through the causal lens and clarify the challenges brought by obstacles. Then in the face of the confounding bias of obstacles, we let the agent learn counterfactual policy conditioned on its original intent so that it can gain experiences of counterfactual situations and make better decisions.

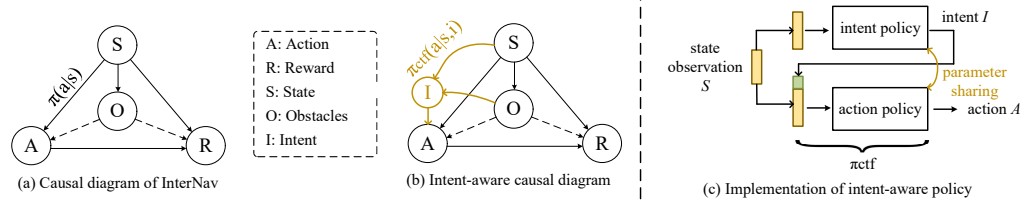

Figure 2: Causal diagram of proposed SCM. The part in gold of (b) denotes the causal entity of the counterfactual policy. The action policy $\pi$ shares the same architecture with the intent policy so that they can receive the state observation in the same form.

## 3.1 Structural Modelling

We first formulate the underlying problem as a time-discrete Markov Decision Process (MDP) defined by a tuple $\langle S, A, T, R, \gamma \rangle$, where $S$ is the state space, $A$ is the action space, $T : S \times A \times S \rightarrow [0,1]$ is a transition function, $R$ is a reward function, and $\gamma \in [0,1]$ is the discount factor. At each step $t$, the agent observes state $s_t = (m_t, p_t)$ representing an egocentric visual observation and a goal coordinate, executes an action $a_t \in A$, and receives a reward $r_t$. Here, the action space $A = \{$*MoveAhead, RotateLeft, RotateRight, LookUp, LookDown, PushAhead, PushBack, PushLeft, PushRight, PickUp, Drop, Done*$\}$.

We then analyze the above MDP with the form of Structural Causal Model (SCM, see Figure 2). Formally, the SCM $M$ is defined by a tuple $\langle O, V, F, P(u) \rangle$, where $V = S \cup A \cup R$ is a set of observable variables, $O$ denotes the obstacles that hinder the navigation of agents, $F = \{f_s, f_a, f_r\}$ is the set of structural functions relative to $V$, and $P(o)$ is the probability distribution over obstacles which is unobserved in InterNav. In particular, objects that currently serve as obstacles are decided by the environmental state such as the layout of surroundings (S→O). At step $t$, action $a_t = f_a(s_t, o_t)$ is decided by the agent based on state observation and its recognition of obstacles (S→A←O). Practically, a stochastic policy $\pi(a_t|s_t) : S \times A \rightarrow [0,1]$ is learned to cover the process. Note that although $S$ denotes the state encompassing all information of the environment, it causally influences agent's actions in a partially observable way through visual observations. Reward $r_t = f_r(s_t, a_t, o_t)$ is returned by the environment based on state, action and obstacles to which the action is applied (S,A,O→R). The current state $s_t = f_s(s_{t-1}, a_{t-1})$ is decided by the state and action at last time step. The goal of the agent is to learn a policy $\pi^* = argmax V_\pi(s_t)$ that maximizes the expected cumulative reward:

$$V_\pi(s_t) = \mathbb{E}[\sum_{k=0}^{\infty} \gamma^k r_{t+k+1}|s_t] \tag{1}$$

which is formulated in the form of state-value function $V_\pi$ and $\gamma$ is the discount factor.

Through interaction with the environment, the agent performs intervention $do(A = a)$, that actively sets $A$ as $a$ rather than passively observes when $A$ appears to be $a$, and learns from the effect which can be written probabilistically as $P(R = r|do(A = a), S = s)$. Since obstacles influence both the decision of actions and the generation of rewards (A←O→R, see dashed arrows in Figure 2(a)), the causal relation between A and R is confounded such that $P(r|do(a)) \neq P(r|a)$, to which the *confounding bias* refers [25]. During RL training, the model estimates spurious value of actions (in the form of reward) due to the confounding bias which may result in sub-optimal policy. For example, frequent encounters with heavy obstacles (e.g. chairs, tables) make the agent predict a higher value of pushing them than picking them up. The agent can get familiar with limited distribution $P(o)$ with extensive training. However, without considering the causalities concerning obstacles (A←O→R), it's hard to generalize to unseen environments or large-scale datasets.

## 3.2 Counterfactual Decision-making

The underlying confounding bias can be tackled with the ability of counterfactual decision-making. A counterfactual effect $P(R_{A=a}|A = a')$ of taking action can be read as "The reward the agent

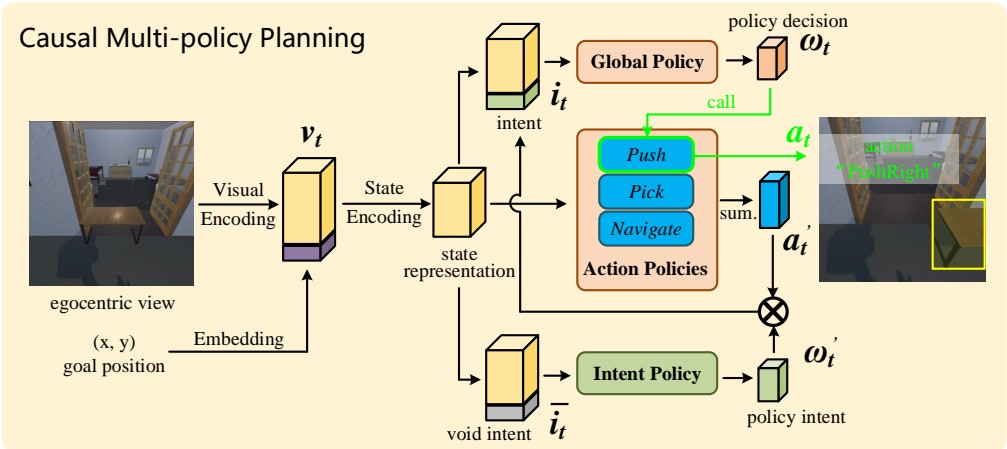

Figure 3: Model overview. Our model CaMP is composed of a global policy, three action policies, and an intent policy. Given an observation, the intent policy predicts a policy $\omega'_t$ and low-level policies predict an action $a'_t$, which is then weighted by the policy distribution as the integrated intent $i_t$ (equation (7)). Finally the global policy decides the policy $\omega_t$ based on the state representation and the intent, and the chosen policy executes the action $a_t$.

would obtain had it taken action $A = a$ (contrary to the fact), given $A = a'$", which is imaginary thus usually not estimable from data. However, Bareinboim et al. [3] show that the above counterfactual quantity can be estimated if the action is interrupted before its decision is executed (ETT, Effect of Treatment on the Treated). We call the decision before execution as agent's intent $I = i_t = f_i(s_t, o_t)$ which shares the same causal parents of action $A$. Our goal is to learn a counterfactual policy $\pi^*_{ctf} = argmax V_{\pi_{ctf}}(s'_t)$ (see Figure 2(b)) that maximizes the long-term value conditioned on the intent-specific state $s'_t = (s_t, i_t)$:

$$V_{\pi_{ctf}}(s'_t) = \mathbb{E}[\sum_{k=0}^{\infty} \gamma^k r_{t+k+1} | s_t, i_t] \tag{2}$$

The action distribution of the counterfactual policy $P(a_t | i_t) = \pi_{cft}(s_t, i_t)$ is posterior to the intent distribution $P(i_t) = \pi(s_t)$. Thus the agent can obtain both experimental experiences (ones the original policy "would have collected" when $a_t = i_t$) and counterfactual experiences (when $a_t \neq i_t$) that explore external strategies. The intent also provides context about the obstacles due to their causal relation. Thus the counterfactual policy obtains more value than standard policy [41]:

$$V_{\pi^*_{ctf}}(s_t, i_t) \geq V_{\pi^*}(s_t) \tag{3}$$

where equality does not hold when unobserved confounders (obstacles) exist. Intuitively, an agent that explores different counterfactual situations (e.g. "what if I push/pick up the box instead of bypassing it?") has a better understanding of the expected value of the current state, compared to the agent taking actions out of intuition. We illustrate the idea of how we implement a counterfactual policy in Figure 2(c).

## 4 The Proposed Solution

In this section, we first introduce a hierarchical framework for multiple policies to cooperate. Then we implement a global policy with the ability of counterfactual decision-making which plans three action policies based on the integrated intent of the model. The model architecture is shown in Figure 3.

### 4.1 Hierarchical Policy Framework

Essentially, we reduce the task complexity following a prior understanding that the InterNav task is temporally separable. 3 action policies are defined in Table 1 to split the action space. Policy

Table 1: Descriptions of action policies

| Policy | Action space | Objective |
|---|---|---|
| Navigate | *MoveAhead, RotateLeft, Rotate-Right, LookUp, LookDown, Done* | Navigating to the goal point within a distance of 0.2m. |
| Push | *PushAhead, PushBack, PushLeft, PushRight, Done* | Moving obstacles until there is a reachable path towards the goal or the existing path gets shorter. |
| Pick | *PickUp, Drop, RotateLeft, Rotate-Right, Done* | Picking and dropping small obstacles until there is a reachable path towards the goal or the existing path gets shorter. |

*Navigate* focuses on the ability of point navigation without considering the environment dynamics. While *Push* and *Pick* policies aim for the ability of object manipulation according to the goal and agent position. Formally, assume action policy $\pi_\omega$ is called at step $t$ mapping the transition from state observation $s_t$ to the distribution of action $a_t$:

$$P(a_t) = \pi_\omega(s_t, h_{t-1}), a_t \in \mathcal{A}_\omega \tag{4}$$

where $h_{t-1}, \mathcal{A}_\omega$ denotes the recurrent hidden state and the action space of policy $\pi_\omega$. In our model, $s_t$ is the concatenation of goal embedding and visual features extracted with a convolutional neural network. $\pi_\omega$ is implemented with a GRU and two linear layers for both policy (actor) and value (critic). We pretrain each action policy with an auxiliary task designed according to their objectives.

With the short-term abilities of action policies, a global policy $\pi_\Omega$ aims to plan a sequence of policies $\omega_t \in \Omega = \{$*Navigate, Push, Pick, Done*$\}$ for the whole task completion. We let the agent follow the *Navigate* policy by default and the global policy decides whether to interact during the long horizon navigation. Once an interaction policy is called, it does not return the control until the sub-task termination (output *Done*). $\pi_\Omega$ shares the same model architecture with $\pi_\omega$.

## 4.2 Intent-aware policy planning

Since the global policy plans when to interact with obstacles, it's crucial to learn action-outcome causality (e.g. whether calling *push* policy results in a clear path). Moreover, unlike low-level policies, the global policy does not receive direct rewards for its decisions during training, which makes their causal relation harder to learn. Thus we introduce a counterfactual global policy which makes plans based on the theory in section 3.2, in order to enable more effective policy learning. The intent of the agent is combined with its observation as the intent-specific state that is fed to the global policy. Instead of planning policies based on its own intent of policy, $\pi_\Omega$ is provided with the integrated intent of basic actions $i_t \in \mathcal{A}$:

$$P(\omega_t) = \pi_\Omega(s_t, P(i_t), h_{t-1}) \tag{5}$$

$$P(\omega_t') = \pi_\Omega'(s_t, P(\bar{i}_t), h_{t-1}) \tag{6}$$

$$P(i_t) = \sum_{\omega^j \in \Omega} \pi_{\omega^j}(s_t, h_{t-1}) \cdot P(\omega_t' = \omega^j) \tag{7}$$

where $\pi_\Omega'$ is the *intent policy* that duplicates the decision-making of global policy and $\bar{i}_t$ is a plane vector denoting an intent without any inclination. Each action intent (zero-padded to the same size) produced by policy $\pi_{\omega^j}$ is weighted by the probability that the decision of $\pi_\Omega'$ is $\omega^j$. Note that $P(i_t)$ is encoded with a linear function and concatenated to the observation $v_t$ as the input of GRU. By providing information of the intents of low-level policies, the new context bridges the relation between actions and rewards for the global policy.

We train the model with Proximal Policy Optimization (PPO) [28] algorithm which updates the model parameters for $k$ iterations with a rollout of data. To ensure that the intent represents the decision of the agent, we synchronize the parameters of two networks ($\pi_\Omega' \leftarrow \pi_\Omega$) during training. Since the global policy does trials conditioned on the distribution of model intent, similar distributions of actions and intents are more likely to be sampled identically (experimental case, $a_t = i_t$). On the contrary, the distinctive distributions of actions and intents lead to more counterfactual cases ($a_t \neq i_t$), which brings entropy increase for the system. In other words, the agent tends to exploit its internal knowledge about influencing UCs (obstacles) receiving its own intent, while exploring the external space of policy when setting off from the old intent. Since the global policy is becoming distinct from the intent policy after iterations, we set the synchronization frequency as $K$ rollouts to balance the exploration and exploitation of policy learning.

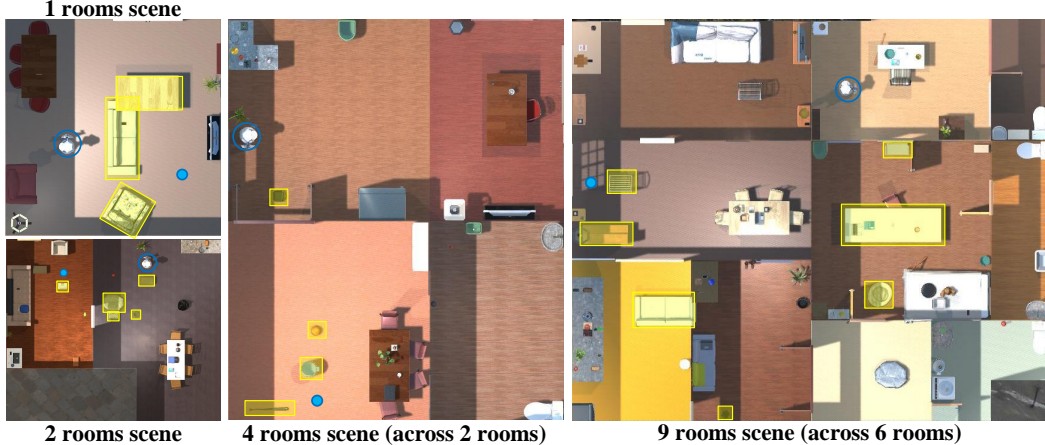

**1 rooms scene**

**2 rooms scene**  **4 rooms scene (across 2 rooms)**  **9 rooms scene (across 6 rooms)**

Figure 4: Top view examples of our dataset. Blue circles denote the starting and target point. Yellow boxes denote the obstacles we spawn on the path.

Table 2: Statistics of proposed dataset split by the number of rooms. The ratio of each split, number of spawned obstacles, initial distance between start and target, and number of times the agent needs to across the room are counted.

| Dataset splits (room num) | ratio | obstacle num | distance(m) | cross-room num |
|---|---|---|---|---|
| 1∼2 | 39.0% | 2.05 | 5.45 | 1.44 |
| 3∼5 | 42.3% | 3.37 | 9.92 | 2.50 |
| 6∼10 | 18.7% | 4.58 | 14.67 | 3.29 |

## 5 Experiments

### 5.1 Experiment Setup

**Data collection.** We perform experiments with the ProcTHOR simulator [14] on our dataset, which contains 600k task episodes in 12k scenes (50 per scene). For each episode, we first randomly set the starting and target points in two different rooms. Second, we calculate all room paths (e.g. room1→room4→room7) and all node paths (e.g. coordinates of start→door3→target). At last we randomly pick obstacle assets (655 in total) at random positions between adjacent nodes until there is no clear path or lengthen the shortest path (50% probability each). The dataset is split by scenes so the models are evaluated in unseen scenes: 11.8k training scenes, 100 validation scenes (5k episodes), and 100 testing scenes (500 episodes). In Table 2 the statistics of our dataset show that the agent is challenged by multiple obstacles, long-distance navigation, and multi-door passing. Figure 4 shows several top-view examples of our dataset.

**Environment settings.** Following the standard settings, we let *MoveAhead* move the agent ahead by 0.25 meters, *RotateRight* and *RotateLeft* change the agent's azimuth angle by ±90 degrees, *LookUp* and *LookDown* rotate the agent's camera elevation angle by ±30 degrees. The *DirectionalPushs* let the agent push (along ±z and ±x axis) the closest visible object with a constant force. The agent takes the *END* to indicate that it has completed an episode.

**Evaluation metrics.** We adopt Success Rate (SR), Final Distance to Target (FDT), and Success weighted by Path Length (SPL). SR is the ratio of successful episodes in total episodes, FDT is the average geodesic distance between the agent and the goal when the episode is finished, and SPL is calculated as $\frac{1}{N}\sum_{n=1}^{N} Suc_n \frac{L_n}{max(P_n, L_n)}$, where $N$ is the total episodes number, $Suc_n$ is the successful indicator of $n$-th episode, $L_n$ is the shortest path length, and $P_n$ is the length of the real path.

In non-interactive navigation, the metric SPL that measures path efficiency is also a usable indicator of time efficiency, since the length of a path is usually proportional to the time it takes to execute the corresponding navigation actions (e.g. move ahead, rotate right). However, a shorter path does not guarantee less time consumption for task completion in InterNav due to the object interaction. For instance, the agent may take the shortest path to the target through extensive interactions which cost a large amount of time and end up less effective than taking a detour path. Thus we evaluate an additional metric STS (Success rate weighted by Time Steps) to measure the time cost of task completion: $STS = \frac{1}{N} \sum_{n=1}^{N} Suc_n \frac{L_n/grid}{TS_n}$, where $L_n$ is the shortest path length, $TS_n$ is the timesteps the agent takes to complete the task, and $grid = 0.25m$ is the unit distance of a step. Thus $L_n/grid$ represents the number of timesteps it takes to navigate to the target by merely moving forward. STS can be regarded as a time-measurement variant of SPL and the score is higher when the agent accomplishes the task with less time. The ideal situation is that the target is directly ahead and there is no need for interaction where STS is 1.

## 5.2 Implementation Details

Our method is implemented and evaluated using the AllenAct [34] framework. The egocentric observation is set as 300*300 RGB and depth images. We first train three action policies then the whole model, utilizing Adam with an initial learning rate of $3 \cdot 10^{-4}$ for policy training and $1 \cdot 10^{-4}$ for joint model training. The hidden size of CNN, GRU is set as 512. The intent embedding size is set as 12 (the same as the intent size and the length of action space). The frequency of synchronization $K$ and epoch number $k$ are set as 3 and 1. All models are trained for 10 million steps with 2 million warm-up steps in single-room scenes (1m for action policies and 1m for joint training) and 8 million in all scenes[1].

**Reward shaping.** We set the reward for our task as three parts: $r = r_{success} + \Delta_{dis} - r_{sp}$, where $r_{success} = 10$ is provided if the agent takes action *Done* meanwhile achieving the goal. $\Delta_{dis}$ represents the difference of geodesic distance between the agent and the goal position, and $r_{sp} = 0.01$ denotes the step penalty.

Three auxiliary tasks are designed as: (1) The *Navigate* policy is pretrained with the PointNav task where the agent moves to the goal point in an obstacle-free environment. The task shares the same reward $r_{nav} = r$ as above. (2) To pretrain the policy of *Push* and *Pick*, the agent is placed aside an obstacle opposite to the goal. It achieves success by taking *Done* when the obstacle is cleared. The reward $r_{inter} = r_{nav} + r_{as} - r_{af}$, where $r_{as}, r_{af}$ denote the successful reward and failed penalty of chosen action (e.g. fail to conduct *Drop* before picking up something). Since effective pushes or picks change the length of the shortest path, $\Delta_{dis}$ also applies to interaction reward.

## 5.3 Baselines

**Random**: This baseline randomly takes actions with the same probability until the agent reaches the target or reaches the maximum number of steps.

**PPO**: We set a common baseline trained with PPO algorithm according to [35]. Similar to our method, it extracts visual features with an encoder and reads the goal position with an embedding layer, then a single policy network predicts actions based on fused features.

**NIE**: We implement the method NIE [39] which predicts the state change of observed objects conditioned on actions with a module in addition to the PPO baseline. The representation of object change is fused with visual features and goal embedding.

**HRL**: To study the effect of our multi-policy framework, we implement a common hierarchical policy supported by AllenAct [34] that shares the same three sub-spaces of action. The difference is that the low-level policies are called at each step, not able to plan actions in a period.

**PPO+intent**: We develop a counterfactual policy based on the PPO baseline according to Figure 2(c) to study the effect of intent-aware decision-making.

---

[1]The project is available at: https://github.com/polkalian/InterNav.

Table 3: Quantitative results.

| Methods | all | | | | N≥ 4 | | | | non-interactive | | | |
|---|---|---|---|---|---|---|---|---|---|---|---|---|
| | SR(%)↑ | FDT(m)↓ | SPL↑ | STS↑ | SR | FDT | SPL | STS | SR | FDT | SPL | STS |
| Random | 1.32 | 7.68 | 0.002 | 0.001 | 0 | 12.29 | 0 | 0 | 2.42 | 7.35 | 0.003 | 0.001 |
| PPO [35] | 42.7 | 4.84 | 0.252 | 0.134 | 23.3 | 7.79 | 0.139 | 0.086 | 51.5 | 3.44 | 0.306 | 0.181 |
| NIE [39] | 52.0 | 3.94 | 0.287 | 0.155 | 37.3 | 6.54 | 0.195 | 0.102 | 58.8 | 3.01 | 0.345 | 0.188 |
| HRL | 43.4 | 4.80 | 0.253 | 0.135 | 24.3 | 7.01 | 0.169 | 0.084 | 51.9 | 3.40 | 0.316 | 0.176 |
| PPO+intent | 54.7 | 3.84 | 0.305 | 0.163 | 38.0 | 6.29 | 0.198 | 0.109 | 70.4 | 2.38 | 0.390 | 0.222 |
| CaMP | **56.3** | **3.67** | **0.327** | **0.177** | **41.4** | **5.76** | **0.231** | **0.121** | **72.3** | **2.05** | **0.407** | **0.236** |

Table 4: Ablation results.

| Methods | all | | | | N≥ 4 | | | | non-interactive | | | |
|---|---|---|---|---|---|---|---|---|---|---|---|---|
| | SR(%)↑ | FDT(m)↓ | SPL↑ | STS | SR | FDT | SPL | STS | SR | FDT | SPL | STS |
| **Sync. variants:** | | | | | | | | | | | | |
| sync. /epoch | 53.0 | 4.16 | **0.316** | 0.160 | 36.7 | 6.95 | **0.196** | 0.107 | 63.4 | 2.53 | 0.375 | 0.219 |
| sync. /rollout | 50.7 | 4.22 | 0.277 | 0.145 | 36.0 | 6.82 | 0.175 | 0.105 | 57.9 | 3.10 | 0.345 | 0.191 |
| sync. /3*rollout | **54.7** | **3.84** | 0.305 | **0.163** | **37.3** | **6.54** | 0.195 | **0.109** | **70.4** | **2.38** | **0.390** | **0.222** |
| sync. /5*rollout | 50.3 | 4.25 | 0.279 | 0.150 | 30.7 | 6.85 | 0.147 | 0.098 | 55.9 | 3.21 | 0.337 | 0.185 |
| **intent variants:** | | | | | | | | | | | | |
| wo/intent | 50.8 | 4.03 | 0.294 | 0.151 | 31.0 | 6.79 | 0.151 | 0.095 | 59.2 | 2.98 | 0.347 | 0.202 |
| test wo/intent | 41.7 | 4.95 | 0.202 | 0.111 | 24.7 | 7.72 | 0.110 | 0.085 | 50.5 | 3.56 | 0.296 | 0.168 |
| recursive intent | 51.4 | 4.06 | 0.292 | 0.156 | 32.7 | 7.04 | 0.159 | 0.096 | 62.8 | 2.61 | 0.353 | 0.215 |
| policy intent | 52.5 | 4.32 | 0.304 | 0.155 | 35.8 | 7.09 | 0.191 | 0.100 | 62.6 | 2.91 | 0.357 | 0.216 |
| integrated intent | **56.3** | **3.67** | **0.327** | **0.177** | **41.4** | **5.76** | **0.231** | **0.121** | **72.3** | **2.05** | **0.407** | **0.236** |

## 5.4 Results Analysis

**Comparison with baselines.** We pick model parameters on the validation set and report their performance on the whole testing set, hard scenes with more rooms ($N \geq 4$), and the splits of episodes that exist clear paths (non-interactive). As quantitatively shown in Table 3, CaMP outperforms all baselines and obtains +13.6/+7.5, +18.1/+9.2, and +4.3/+3.5 improvement in SR, SPL, and STS (all/$N \geq 4$, %) respectively compared to the model PPO. Compared with PPO+intent, CaMP obtains more improvement in hard scenes (+3.4 and +3.3 in SR, SPL) which shows the robustness of multi-policy facing complex environment. Besides, the relatively poorer performances of HRL and Random model reflect the challenge of our dataset. The limited performance in hard scenes shows that effective methods for navigation in plenty of rooms are still in need. The contrary trends across STS and SPL when comparing PPO and HRL on the hard split and the non-interactive split indicate that both path and time efficiency should be measured considering InterNav scenarios.

**Ablation study.** Our ablation study aims to explore the effect of two model designs. First, we change the synchronization frequency of model PPO+intent as 1 epoch, 1 rollout, 3 rollouts, and 5 rollouts. As shown in Table 4, the variant with $K = 3$ gains the relatively best performance, which reflects how the balanced exploration-exploitation benefits policy learning.

Second, we study how the model intent influences performance: We implement (1) CaMP without the intent network as *wo/intent*, (2) CaMP that inferences without the intent network as *test wo/intent*, (3) CaMP provided with the intent recursively generated based on the prior intent as *recursive intent*, and (4) CaMP provided with the intent of the global policy as *policy intent*. Here our model CaMP is referred to as *integrated intent*. As shown in Table 4, the integrated intent effectively improves the decision-making ability of the global policy compared with *wo/intent* (+5.5/+3.3 and +10.4/+8.0 in SR, SPL). Without the information of the intent of action policies, *policy intent* shows less gain effect compared with *integrated intent* (-3.8/-2.3 and -5.6/-4.0 in SR, SPL). The limited performance of *test wo/intent* suggests that the inference process of our model also relies on the intent. The results of *recursive intent* suggest that although the secondary intent still provides incremental information about the system, the counterfactual situation based on the "counterfact" may become less meaningful for learning causality.

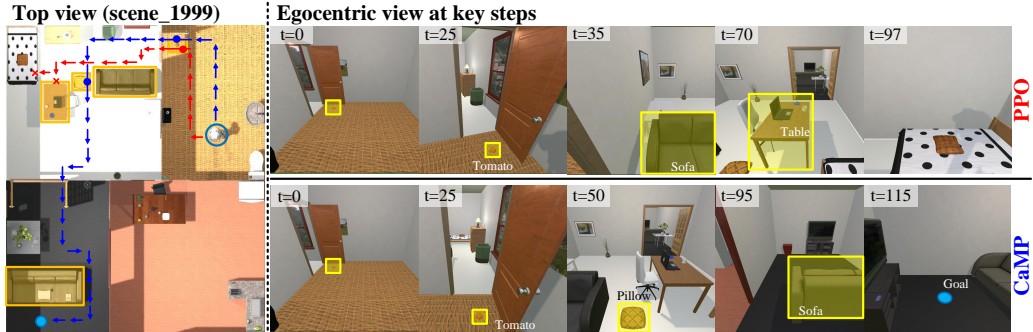

Figure 5: Qualitative cases. The starting and target points are marked by blue circles and obstacles are marked by yellow boxes. The blue arrows and red arrows denote the trajectories from CaMP and PPO, respectively. The nodes and crosses denote successful object interactions and failed ones. Markings in colors are invisible to agents.

**Case study.** We qualitatively compare our model with a baseline PPO on a typical instance in a 4-room scene in Figure 5. The blue trajectory of CaMP shows that the agent successfully removes obstacles of *tomato* and *pillow*, and attempts to bypass an immovable *sofa* to reach the target. While the red trajectory shows that PPO intends to interact with a *table* and a *bed* which are too heavy to be pushed aside and ends up getting stuck in a corner. The comparison suggests that our model can make better decisions on whether and how to interact with obstacles.

Table 5: Obstacle interaction results.

| Methods | IR(m) | ISR(%) | PuSR(%) | PiSR(%) |
|---|---|---|---|---|
| PPO [35] | 0.271 | 23.2 | 23.8 | 21.9 |
| PPO+intent | 0.371 | 27.0 | 25.1 | 28.6 |
| CaMP wo/intent | 0.287 | 20.4 | 19.9 | 23.5 |
| CaMP | 0.402 | 27.6 | 27.1 | 29.8 |

**Study of obstacle interaction.** To study how intent-aware policy learns the obstacle interaction and its outcome, we make a direct evaluation on obstacle interaction of several models in Table 5: (1) Interaction Reward (IR): the reduction of geodesic distance $\Delta_{dis}$ caused by interactions. (2) Interaction Success Rate (ISR): the ratio of taking interactions successfully (a more detailed measurement compared to the overall metric SR measuring task completion). A failed case of interaction is when the agent tries to pick up a heavy table or attempts to push a chair out of sight. We also report the success rate of Push and Pick interactions separately as PuSR (Push Success Rate) and PiSR (Pick Success Rate). The results (comparison between PPO and PPO+intent, CaMP and CaMP wo/intent) show the outcome of our causal method in pursuing the interaction efficacy of reducing the distance to the target (IR) and avoiding invalid actions (ISR). The results on PuSR and PiSR also suggest that successfully pushing an obstacle is more challenging since the obstacle may get stuck and is not movable in all directions.

## 6    Conclusions

In this paper, we tackle the task of Interactive Navigation (InterNav) which requires mastery of long-horizon navigation and object interaction in complex environments. We take a causal view of InterNav and clarify the challenge caused by the confounding bias of obstacles. A Causal Multi-policy Planning (CaMP) model is proposed following the main idea of reducing inefficient exploration with multi-policy and exploring counterfactual interaction by considering the primary intent of model's decision-making. We construct a large-scale dataset of interactive navigation with various obstacles spawned across multiple rooms. Experiments on our dataset illustrate the effectiveness of our method and support our causal modeling of InterNav.

## Acknowledgment

This work is supported by the National Key Research and Development Project of New Generation Artificial Intelligence of China, Grant Number: 2018AAA0102504. This work is in part by Beijing Natural Science Foundation under Grant JQ22012, and in part by the National Natural Science Foundation of China under Grant 62125207.

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

# A Appendix

## A.1 Proof of equation (3)

For a state $s_t$ and intent $i_t$ at step $t$, the optimal counterfactual policy $\pi_{ctf}^* = argmaxV_{\pi_{ctf}}(s_t, i_t)$, and the optimal standard policy $\pi^* = argmaxV_\pi(s_t)$, we have $V_{\pi_{ctf}^*}(s_t, i_t) \geq V_{\pi^*}(s_t, i_t)$. For a standard optimal policy, since $\pi^*$ executes its intent $i_t$ as an optimal action $a_t'$ without re-consideration, we have:

$$a_t' = argmax\mathbb{E}[\sum_{k=0}^{\infty} \gamma^k r_{t+k+1}|s_t, a_t] \tag{8}$$

$$
\begin{aligned}
V_{\pi_{ctf}^*}(s_t, i_t) \geq V_{\pi^*}(s_t, i_t) &= V_{\pi^*}(s_t, a_t')\\
&= \mathbb{E}[\sum_{k=0}^{\infty} \gamma^k r_{t+k+1}|s_t, a_t']\\
&\geq \sum_{a_t \in A} \pi^*(a_t|s_t)\mathbb{E}[\sum_{k=0}^{\infty} \gamma^k r_{t+k+1}|s_t, a_t]\\
&= V_{\pi^*}(s_t)
\end{aligned}
\tag{9}
$$

Thus considering the information of its own intent, the agent can learn a counterfactual policy gaining more value.

## A.2 Impact of different intents on policy exploration

We train our model with PPO [28] which encourages the exploration with an entropy loss $L_t^S = c_2 S[\pi] = -c_2 \sum_{a_t \in A} P(a_t)logP(a_t)$ where $c_2 < 0$ is the coefficient. Since our agent does trials conditioned on its intent, it maximizes the conditional entropy $H(A|I) = -\sum_{a_t \in A} P(a_t|i_t)logP(a_t|i_t)$. We are curious about how this conditional exploration can make the new policy distribution different from the conditional distribution (intent). To study the question and also how the similarity between intent and action affects policy exploration, we evaluate the mean KL divergence between the current and old policy distributions under several settings throughout the training. In particular, we vary the frequency of synchronizing the action policy and the intent policy as 1 epoch, 3 epochs, 6 epochs, 9 epochs, 12 epochs, 15 epochs, and without intent. In Figure 6, we report the mean KL divergence between the distributions of current policy and different old policy (from 1 epoch behind to 15 epochs behind). The trend of each curve shows that the counterfactual information of intent encourages the agent to explore new strategies. The comparison between curves shows that the more different actions and intents, the more the model tends to explore new policies rather than exploit the old policy. Thus the frequency of synchronization can be utilized to adjust how much the agent explores new policies under counterfactual situations.

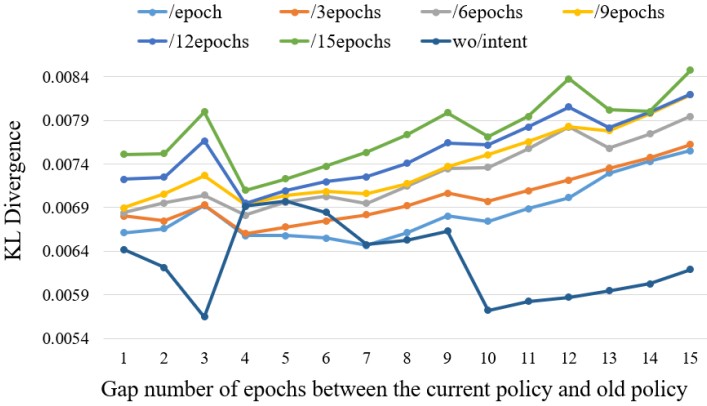

Figure 6: Mean KL divergence of the current and old policy distributions.

