# OpenReview forum: "CaMP: Causal Multi-policy Planning for Interactive Navigation in  Multi-room Scenes"
_NeurIPS.cc/2023/Conference — NeurIPS 2023 poster_

### Official Review · Reviewer_6UQq · 2023-07-06

**Soundness:** 3 good
**Presentation:** 3 good
**Contribution:** 4 excellent
**Rating:** 7
**Confidence:** 4

**Summary:**

This paper introduces the multi-room interactive navigation problem and proposes a novel model that is motivated by counterfactual reasoning. In particular, the paper posits that obstacle objects serve as a confounding factor when understanding the relationship between actions taken and the outcomes observed / reward received. To address this, a counterfactual reasoning based model is proposed, which explicitly encourages exploring actions outside the distribution predicted by the policy (i.e., what would happen if I did action X instead of Y?). The proposed model is hierarchical in nature, with low-level policies for navigation, picking, and pushing skills, and higher-level policies for selecting a skill to execute. The higher-level policy is conditioned on an intent, specifying where the policy is likely navigating to next. This is expected to allow the policy to account for the intent and explore actions that go against the intent.  Results on the ProcThor simulated dataset demonstrate the superiority of the proposed policy over alternative baselines.

**Strengths:**

* The idea of performing counterfactual reasoning is interesting and novel in the context of embodied AI.
* The problem setting proposed is a good extension to prior work on single-room interactive navigation and is valuable for the community to work on.
* The paper clarity is good, but it makes assumption about how knowledgable the reader is with causal inference (see weaknesses).
* The experiments are well designed and ablation studies convey useful information to understand the overall model. The proposed model also performs much better than reasonable baselines.

**Weaknesses:**

# Post-rebuttal comments
* The authors have sufficiently addressed my concerns and provided new experiments to quantify the improvement in interaction-ability. I'm happy to raise my rating to accept.

--------------------------------------------------------------------------------------------------
# Paper writing clarity
* The paper writing clarity can be improved quite a bit with regards to causal inference. Since this is a relatively new topic to the embodied AI space, most readers may be unfamiliar with the topic and jargon like "counterfactual reasoning", "confounding factors", "structural causal model", "structural functions", etc. also, L129 was not obvious from my first reading (i.e., the difference between do(A) and A).
* More clarity can be provided about what the counterfactual situations here are (e.g., more examples like L131-133).
* In Figure 3, is the entire model differentiable? For example, are gradients from the loss propagated through to intent predictions w_t^{'} and a_t^{'}?
* In the experiments, an explicit connection should be made to how the proposed model is using counterfactual reasoning and how it addresses the issue of confounding factors.

# State vs. obstacles as confounding factors
The idea of treating obstacles as confounding variables makes sense. But at a high level, isn't the state variable itself a confounding factor? What is the value is isolating only obstacles here?

# Experiment section can be improved
* Error bars are missing in Tables 3 and 4. It will be useful to have results from training and evaluating on multiple seeds, especially for the top-3 methods (NIE, PPO+intent, CaMP). Similar issue for Table 4.
* In Table 4, why is "sync. /3*rollout" worse than "integrated intent"? Aren't they both the same models?
* In Table 4, why is "sync. /epoch" worse than "wo/intent"? If CaMP is synchronized frequently, I would expect the intent to be selected as the action itself, leaving little to no room for counterfactual exploration. So "sync. /epoch" should match "wo/intent", right?
* There is no analysis on how well each models interact with the "relevant" obstacles, i.e., obstacles that lie along the shortest path and can be moved. This can be measured via precision and recall metrics.
    * Precision = fraction of interacted objects that are "relevant" obstacles
    * Recall = fraction of "relevant" obstacles interacted with

**Questions:**

Kindly address the weaknesses stated above.

**Limitations:**

No limitations have been discussed.

---

> ### Author Rebuttal · Authors · 2023-08-09
>
> Thanks to the reviewer for the appreciation and suggestions for our work. We address the concerns in the following lines.
>
> Q1. Paper writing clarity. The reviewer questions about "the difference between do(A) and A", "what the counterfactual situations here are (e.g., more examples like L131-133)", and "In Figure 3, is the entire model differentiable?"
>
> A1. Thanks for the suggestion and we will revise our paper to further elaborate on the causal preliminaries. Our replies to the questions are as below:
>
> * Do-calculus $do(A=a)$ is an intervention operation that actively sets the variable $A$ as $a$ in the system while keeping the rest of the system unchanged, and the postintervention distribution $P(R|do(A=a))$ is obtained. On the contrary, the conditional distribution $P(R|A=a)$ is observed passively in the condition when $A$ appears to be $a$. Thus the actions executed by embodied agents can naturally be regarded as interventions do(A) to the environment. Formally in our case (Figure 2(a)), distribution $P(R=r|do(a))=\sum_{x} P(r|a,s)P(s)$, while $P(R=r|a)=\sum_{x} P(r|a,s)P(s|a)$. Since the agent decides the action $a$ based on the observation of state $s$, the distributions of $a$ and $s$ are not independent, which results in $P(s)\neq P(s|a)$ and $P(R=r|do(A=a))\neq P(R=r|A=a)$.
> * The counterfactual situation we discuss $R_{A=a}|A=i$ can be understood as the result the agent would obtain had it taken action $a$, given that it intends to take $i$. For example, in the initial stage of training, when encountering a table the agent may intend to bypass the obstacle since it lacks interaction skills and interactions empirically lead to low rewards. With the help of counterfactual policy, the agent may explore the counterfactual situation when it tries to push the table aside given its intent is to rotate right. Once the interaction successfully clears the path, the agent will obtain a relatively higher reward than navigation, which becomes a valuable experience for the agent to learn interactive strategy.
> * The entire model is differentiable, because the integrated intent $i_t$ is obtained with
> the weighted-sum of low-level action distribution. However, we do not update the intent network with gradients from the loss. Instead the parameters of the intent network are updated by synchronization with the master policy.
>
> Q2. State vs. obstacles as confounding factors. The reviewer questions that "But at a high level, isn't the state variable itself a confounding factor? What is the value is isolating only obstacles here?"
>
> A2. We address obstacles in isolation since they are crucial Unobserved Confounders (UC) with negative impact while the distribution of state is observable through visual observations, although state $S$ indeed is a confounder of causality $A \rightarrow R$. In essence, the agent aims to learn and optimize the state-specific causal effect $E[R|do(A=a),S=s]$ (L127), which represents the reward expectation when the agent takes action $a$ observing state as $s$. Therefore the causality $S \rightarrow R$ in Figure 2 is also useful for policy learning and should be considered. However, the distribution of obstacles can not be measured by policy $\pi(s)$, which is harmful to the learning of effective policy.
>
> Q3. Experiment issues. The reviewer argues that "Error bars are missing in Tables 3 and 4.", "There is no analysis on how well each models interact with the "relevant" obstacles", and questions about the model performances.
>
> A3. Thanks for the concerns and we address them by items as follows:
> * We report the performances of models (in Table 3,4) by averaging their scores on 5 tests with the same random seed. We now evaluate several methods on **multiple seeds** in the table below (in the form of *mean* $\pm$ *variance*). The results show that the methods obtain stable performances on multiple seeds.
> * In the ablation study, we vary the sync frequency on model "ppo+intent" (L264) in order to study the effect on counterfactual policy and remove the influence of applying multi-policies. Thus in Table 4, "sync./3*rollout" corresponds to "ppo+intent" while "integrated intent" is identical to CaMP.
> * According to the clarification above, the "sync. variants" are modified based on "ppo+intent" while the "intent variants" are modified based on CaMP in Table 4, which makes the performance of "sync./epoch" relatively worse than "wo/intent".
> * In InterNav, the agent attempts to interact with the "relevant" obstacles for the purpose of navigating more efficiently along a shorter path. Thus how well the model interacts with obstacles actually lies in how much the interactions shorten the path. To that end, we apply SPL metric to measure the length of the path the agent takes to complete the task.
>
> Furthermore, to answer the question "Does the shortest path mean the most efficient navigation?", we consider an example where a large-size table lies in front of the agent on the shortest path, with an apple on the right side of the table. Obviously, moving the table aside is harder than picking up the apple and may be less efficient. Thus we additionally evaluate a metric STS (Success rate weighted by Time Steps, see our reply to *reviewer HVkt* for more details) to analyze the time efficiency of task completion in the table below. The results on SPL and STS show that our method obtains better path and time efficiency compared with the baselines.
>
> |Methods|SR (%)|SPL($_{\pm e-5}$)|FDT($_{\pm e-3}$)|STS($_{\pm e-5}$)|
> |:----:|:----:|:----:|:----:|:----:|
> |PPO|42.3$_{\pm 0.57}$|0.249$_{\pm 1.98}$|4.86$_{\pm 1.56}$|0.139$_{\pm 0.87}$|
> |NIE|51.3$_{\pm 1.01}$|0.290$_{\pm 2.45}$|3.86$_{\pm 4.22}$|0.157$_{\pm 2.22}$|
> |PPO+intent|53.0$_{\pm 1.93}$|0.296$_{\pm 2.20}$|3.80$_{\pm 1.71}$|0.163$_{\pm 1.14}$|
> |CaMP|**56.7**$_{\pm 1.19}$|**0.309**$_{\pm 2.75}$|**3.71**$_{\pm 1.97}$|**0.176**$_{\pm 1.02}$|

---

> > ### Comment · Reviewer_6UQq · 2023-08-13
> > **Reviewer response to rebuttal**
> >
> > I thank the authors for their detailed and helpful rebuttal responses. Quite a few of my concerns are alleviated. I have one follow-up question, and I will decide my final rating based on the above responses and responses to the follow-up question.
> >
> > ## Detailed comments to each response
> > * Q1 - thank you for the clarification. Please incorporate these in the paper to strengthen further.
> > * Q2 - I understand the point that the authors are making. The problem here is using “state” to refer to the agent’s observations is a misnomer. By definition, the state must encompass everything there is to know about the world and the agent, and includes the obstacles as well. In this work, “state” is referred to as the limited part of the state called “observations”, and that may not include the obstacle information. It would be great if this can be clarified in the final paper too.
> > * Q3a - Thanks for the random seed experiments. Please bold results based on significance testing. It looks like only SR and STS improvements are statistically significant relative to NIE / PPO+intent. SPL and FDT are comparable. I expect the gains to be “significant “for the harder case.
> > * Q3b,c - Good point. My bad, thanks for clarifying.
> > * Q3d - I agree that the numbers are better with intent (e.g., PPO vs. PPO+intent). I am just wondering if there are more direct metrics that can quantify how well the agent learns to interact with objects due to causality. E.g., the PPO+intent agent does not try to lift heavy objects and moves them instead — unlike PPO, the PPO+intent agent avoids interactive with large and heavy objects and moves around them instead.
> >
> > ## Follow-up question
> > * Could the authors please define more direct metrics that can quantify the outcomes of the proposed model (other than the indirect task-performance metrics)? The authors have motivated the need for causal reasoning, but whether any of these outcomes are achieved is unclear. I have given some examples above in the detailed comments.

---

> > > ### Author Response · Authors · 2023-08-15
> > >
> > > We thank the reviewer for the recognition and further concern. We will incorporate the clarification (Q1) and the results (Q3) of our rebuttal in the final paper. Our response to the further comment and the follow-up question is as below.
> > >
> > > * **Response to further comment on Q2:** Thanks for pointing that out. It's worth clarifying in the final paper that the policy $\pi(a_t|s_t)$ takes in the observation of the state rather than the whole state information. In the causal model we define in Figure 2, the node $S$ denotes the state encompassing all information of the environment, which includes all potential obstacle information (node $O$). However, the state and obstacles causally influence agent's actions in a partially observable way through visual observations (i.e. $S \rightarrow A$, $O \rightarrow A$).  And we focus on addressing the crucial UC (obstacles) instead of all unobserved states in our work.
> > >
> > > * **Response to the follow-up question:** We agree that the metrics of overall performance (SR, SPL) are indirect and not enough for measuring the interaction outcome of our method. Contrary to acting intuitively, our causal agent learns to take "wiser" interactions than its intuition.  Thus for verification, we evaluate PPO and PPO+intent on several additional metrics in the table below: (1) **Interaction Reward (IR)**: the reduction of geodesic distance $\Delta_{dis}$  caused by interactions. A case of high SR is that the agent pushes a table aside (rather than forward) to clear the path, knowing it will help complete the task better. (2) **Interaction Success Rate (ISR)**: the ratio of taking interactions successfully (a more detailed metric compared to the overall metric SR measuring the task completion). A failed case of interaction is that the agent tries to pick up a heavy table or attempts to push a chair far away, without realizing the feasible interactions in the current situation. We also report the success rate of *Push* and *Pick* interactions separately as **PuSR** (*Push* Success Rate) and **PiSR** (*Pick* Success Rate). Through the metrics above, we measure the outcome of our causal method in pursuing the interaction efficacy of reducing the distance to the target (IR) and avoiding invalid actions (ISR). The results show that compared with a standard policy, our intent-aware policy gains a stronger ability to take effective and appropriate interactions.
> > >
> > > |Methods|IR($_{\pm e-5}$)|ISR(%)|PuSR(%)|PiSR(%)|
> > > |:----:|:----:|:----:|:----:|:----:|
> > > |PPO|0.271$_{\pm3.54}$|23.2$_{\pm1.12}$|23.8$_{\pm1.18}$|21.9$_{\pm0.97}$|
> > > |PPO+intent|0.371$_{\pm2.28}$|27.0$_{\pm1.27}$|25.1$_{\pm1.44}$|28.6$_{\pm1.22}$|

---

> > > > ### Comment · Reviewer_6UQq · 2023-08-21
> > > > **Reviewer response to rebuttal (part 2)**
> > > >
> > > > I thank the authors for agreeing to incorporate the feedback and for further responses.
> > > >
> > > > Regarding the follow-up question, I am happy with the additional metrics introduced by the authors. These directly capture what the intent-aware model is doing and also suggests some scope for improvement in the PuSR metric (i.e., the model learns to capture pick interactions better than push interactions). I'm looking forward to the updated version of the paper with more discussions on these results.
> > > >
> > > > Considering the rebuttal responses and further discussions, I'm happy to increase my rating to Accept (7). This is an interesting paper and is valuable for the community moving forward.

---

### Official Review · Reviewer_KGF8 · 2023-07-07

**Soundness:** 3 good
**Presentation:** 2 fair
**Contribution:** 3 good
**Rating:** 6
**Confidence:** 3

**Summary:**

Broadly, the paper tackles the Interactive Navigation task: navigating to a goal and interacting with obstacles as necessary, e.g. pushing a chair out of the way.

They use the ProcTHOR simulator with 12k multi-room scenes and generate navigation episodes that are suitably cluttered with obstacles. Their embodiment is abstract, with a discrete high-level action space (e.g. PushLeft, PickUp, Drop, RotateLeft).

Their approach is a hierarchical model, with three pre-trained low-level "action policies" (skills): navigate, push, and pick. A master policy sequences the three. Specifically, the navigation policy is run by default; the master policy can interrupt it by invoking an interaction policy (push or pick); the interaction policy then runs exclusively until it self-terminates, returning control to the master policy.

Their main contribution is the addition of an intent policy; they claim this helps the master policy make better decisions in the presence of obstacles. I'll describe this more below.

They evaluate their approach against several baselines:
* random actions
* a monolithic (non-hierarchical) sensors-to-actions policy trained with end-to-end RL, labeled as PPO
* a prior approach labeled as NIE that aims to predict state change of observed objects
* a hierarchical policy that uses the same nav/push/pick action subspaces as CaMP but differs in how they are combined
* a monolithic sensors-to-actions policy with the addition of CaMP's intent policy

They also study ablations related to the intent policy.

I'll now describe the intent policy in detail, with the caveat that I'm looking for further clarification from the authors here (see Questions section).

The architecture of the intent and master policy are mostly identical; their inputs include goal embedding and extracted visual features and they feature recurrent units.

At timestep t, the intent policy outputs w'.t, a distribution over the three action policies, essentially choosing between push/pick/navigate. Meanwhile, each action policy j ouputs a'.t^j, a distribution over discrete actions. Finally, the intent is the sum of all a'.t weighted by w'.t. For example, if the intent policy on a given step is biased towards push and the push action policy is biased towards the PushLeft action, the overall intent will be biased towards PushLeft.

This intent is fed to the master policy alongside observations (together, "intent-specific state"), which chooses an action policy (either allowing the nav policy to continue, or interrupting by invoking push or pick).

The intent policy and master policy generally share parameters and generally behave similarly, except the intent policy is synced from the master policy only periodically during training, such that in practice the intent will differ from the taken action some of the time.

They describe this intent-informed policy as "exploring counterfactuals", and a component here is learning a value function for the above-mentioned intent-specific state.

**Strengths:**

The InterNav task is a challenging, relevant task for Embodied AI.

The task, scene/episode dataset, and model architecture are clearly communicated in figures and text, with minor exceptions noted below.

The scenes and episodes appear diverse and high-quality.

Baselines and ablation study are rigorous.

The authors show strong performance against their baselines.

**Weaknesses:**

There's ambiguity on some paper details. I'll list bullets here; see my questions for more details:

= action space and positioning of objects

= object dynamics in ProcTHOR

= details of invoking and terminating interaction subtasks

= learning for the intent policy parameters

The paper doesn't discuss sim-to-real transfer or otherwise discuss how CaMP might be applied in a real-world setting.

The biggest weakness is that the intuition and conceptual value of the intent component aren't clearly articulated. For details, see my questions below.

**Questions:**

In Figure 5 and in your video at 0:33, we see the agent making only 90-degree turns. However, the PPO baseline visualized in Figure 5 shows other turn angles. Can you explain this? Is your CaMP agent trained using a more restricted action space (versus action space in baselines)? If so, does this make your comparison results to baselines less clear? I would speculate that restriction to 90-degree orientations would simplify learning particularly in your scenes, which appear to be mostly axis-aligned (Figure 4; walls and most furniture). And a related question: 5.1 mentions "adjacent nodes". Is there a discrete structure to the position of objects, e.g. a grid?

Can you explain a bit about the object dynamics of ProcTHOR, specifically, how does the simulator decide which objects are nonmovable, pushable, or pickable? And can you speculate on how the trained CaMP policy is able to predict this? I.e. is it overfitting to a specific object set, generalizing based on apparent object size in the image sensors, or something else?

>Once an interaction policy is called, it does not return the control until the sub-task termination (output Done).

Can you elaborate on how this works? For example, does the master policy include a recurrent unit, and does this unit receive continuous observations even while an interaction sub-task is running? Related: it would help to modify Figure 3. As-is, Figure 3 suggests that the master policy is free to select any action policy on any step.

>To ensure that the intent represents the decision of the agent, we synchronize the parameters of two networks (π'Ω ← πΩ) during training.

In between these syncs, is the intent policy being updated during the learning phase of PPO, or is it frozen?

Can you discuss sim-to-real? How might CaMP be used in a real-world setting?

Related to sim2real, can you also address one specific concern? First, a few assumptions:

= you wish to keep the same hierarchical architecture

= you could build low-level push/pick/nav policies that work on a real robot

= you require most training of the master policy to happen in simulation due to the required scale of experience-collection

My specific sim2real concern: from 0:29 in the video, it appears that the push subtask doesn't move the agent or change its view (it's a "magic push"). The agent isn't required to approach the object in a certain way or move itself to effectively push the object. So, I speculate that this makes it remarkably easy for the master policy to chain navigation and pushing: the navigation policy essentially resumes after push completion from exactly its last pose, almost as if the obstacle were teleported away. Chaining real-world nav and push subtasks in this way would likely fail due to the push task moving the agent.

I'd like greater clarity in section 3.2, on counterfactual decision-making and the basis for your intent policy and overall causal policy. A few specific questions:

>Trained based on its own intent, the agent can obtain both experimental experiences (when a.t = i.t) and counterfactual experiences
(when a/t != i.t), boosting the exploration of new strategies

Can you elaborate? How does the causal policy architecture "boost" this exploration? How does this produce better exploration than, say, simply tuning temperature on a stochastic policy?

>The intent also provides context about the obstacles due to their causal relation

Can you elaborate? Can you experimentally verify/quantify what context the intent is providing about obstacles?

>Intuitively, an agent that explores different counterfactual situations (e.g. "what if I push/pick up the box instead of bypassing it?") has a better understanding of the expected value of the current state, compared to the agent taking actions out of intuition.

It seems to me that a stochastic policy will try pushing, picking, and bypassing the box many times over the course of large-scale training and thus get this understanding of the expected value. What is the essential difference with your causal policy?

**Limitations:**

No concerns here

---

> ### Author Rebuttal · Authors · 2023-08-05
>
> We appreciate the detailed questions and reply to them in the following lines.
>
> Q1. Action space and positioning of objects.
>
> A1. Our CaMP agent is trained using exactly the same action space as other baselines. In particular, the agent makes only 90-degree turns when taking *RotateRight* and *RotateLeft*. The turning angles shown in Figure 5 are just for visualization by combining the trajectories of several adjacent steps. In section 5.1, we mention "adjacent nodes" referring to different doors on the path of cross-room navigation. As for the position of objects, all obstacles are spawned on the reachable points of agents, which are distributed on a grid network with a grid size of 0.25m.
>
> Q2. Object dynamics in ProcTHOR.
>
> A2. The interactive attributes of objects are decided according to their object categories (e.g. all tables are unmovable) in the ProcTHOR simulator. We speculate that CaMP policy learns about those attributes by generalizing based on both visual appearance from image and depth sensors and prior experience of interaction.
>
> Q3. Details of invoking and terminating interaction subtasks.
>
> A3. The master policy consists of a GRU and a MLP, and keeps receiving observations when an interaction sub-task is running, yet the output of the master policy will not influence the behavior of the agent. In Figure 3 we use green arrows to denote the call and control procedure of multi-policy and we will modify it to further clarify the transition of control.
>
> Q4. Learning for the intent policy parameters.
>
> A4. We keep the parameters of the intent policy frozen in between syncs since the intent produced by the intent network is just for replicating the decision of the master policy.
>
> Q5. Sim-to-real transfer and application in a real-world setting.
>
> A5. We appreciate the decent concern and agree with the assumptions. It's true that the sim-to-real gap of object interaction supported by ProcTHOR is quite big compared with that of navigation. However, we speculate that the main challenge of applying our method lies in interaction rather than navigation, since object manipulation and interaction dynamics are more complex to learn with limited training data in the real world. Meanwhile, in our hierarchical framework, each call of the nav policy can be regarded as a solo navigation heading toward the target with different starting positions, which reduces the complexity of long-term navigation. Recent development of simulators [1,2] makes the large-scale training for object manipulation with robotic arms possible. Thus we believe it's practical to transfer our method first to manipulation-included, simulated tasks and then to real-world settings to overcome the sim-to-real challenge.
>
> **Q6. Intuition and conceptual value of the intent component**. The reviewer questions about "How does the causal policy architecture "boost" this exploration", "What is the essential difference with your causal policy" compared with a stochastic policy, and "what context the intent is providing about obstacles".
>
> A6. We elaborate our idea of "intent-aware counterfactual exploration" as follows.
>
> * First, counterfactual policy boosts the exploration of new strategies by accumulating new experiences posterior to that from the original policy. During RL training, an experience used for loss calculation is commonly denoted as a tuple $e_t=\left \langle s_t,a_t,r_t,s_{t+1} \right \rangle \in E$, where $r_t,s_{t+1}$ are determined by $s_t$ and $a_t$. Since the counterfactual policy is aware of intent that $s_t^{\prime}=(s_t,i_t)$, experiences from it $E^{\prime}$ include more information than experiences from a standard policy $E$. Moreover, the action distribution follows $P(a_t|i_t)=\pi(s_t,i_t)$, which is posterior to the intent distribution $P(i_t)=\pi(s_t)$. Thus $E^{\prime}$ contains two sets: experimental ones $E_{exp}|\_{a_t=i_t}$ that the original policy "would have collected", and counterfactual ones $E_{ctf}|\_{a_t \neq i_t}$ that are new experiences exploring other strategies.
>
> * Therefore, second, there are three advantages of counterfactual exploration compared with stochastic exploration:1) $s_t^{\prime}$ provides more knowledge about the environment (discussed below). 2) While stochastic experiences are independently distributed, $E_{ctf}$ are collected based on the original strategy, making them more valuable. For example, an agent, who tries to push a box knowing its original intent is to bypass it, may learn that pushing is more effective in this situation since it receives a higher reward. Yet random trials can be hard to correlate and result in inefficient learning. 3) Stochastic experiences are not applicable to on-policy algorithms like PPO since random policy differs too much from the policy we are training. In contrast, counterfactual policy explores new experiences by merely changing the input (add intent) while maintaining the same policy function $\pi$.
>
> * Third, the intent encodes knowledge about obstacles through agent's understanding of the environment. Since the intent shares the same causal parents of action (S $\rightarrow$ I $\leftarrow$ O in Figure 2), the policy network can be regarded as an encoder leveraging its prior experience. Thus intent is a proxy for influencing UC [3].
>
> **References**
>
> [1] Ehsani, Kiana, et al. "Manipulathor: A framework for visual object manipulation." Proceedings of the IEEE/CVF conference on computer vision and pattern recognition. 2021.
>
> [2] Xiang, Fanbo, et al. "Sapien: A simulated part-based interactive environment." Proceedings of the IEEE/CVF Conference on Computer Vision and Pattern Recognition. 2020.
>
> [3] Forney, Andrew, Judea Pearl, and Elias Bareinboim. "Counterfactual data-fusion for online reinforcement learners." International Conference on Machine Learning. PMLR, 2017.

---

> > ### Comment · Reviewer_KGF8 · 2023-08-16
> >
> > Thanks for your detailed rebuttal! I've revised my rating and I have no other questions.

---

> > > ### Author Response · Authors · 2023-08-21
> > >
> > > Thanks for the appreciation and useful suggestions. We'll revise the final version according to the suggestions and our responses.

---

> ### Author Response · Authors · 2023-08-13
>
> We really appreciate the reviewer for the concerns and suggestions for this work. We hope our response has resolved the confusion and questions in the review. If there are any questions or further comments, please let us know and we will try our best to answer them!

---

### Official Review · Reviewer_7DX2 · 2023-07-07

**Soundness:** 2 fair
**Presentation:** 2 fair
**Contribution:** 3 good
**Rating:** 5
**Confidence:** 4

**Summary:**

This paper introduces a causally-inspired hierarchical policy framework for the interactive navigation task in the AI2THOR environment. The framework consists of a master policy, intent policy, and three sub-control policies. The intent policy embeds intuitive intents from the sub-control policies into the master policy, enabling it to make counterfactual decisions. The proposed approach, named CaMP, is evaluated on a newly collected dataset in the ProcTHOR multi-room scenes. The experimental results show that CaMP outperforms the baselines, achieving the best performance on the interactive task.

**Strengths:**

+) The interactive navigation (IN) task is an interesting and challenging task in the field of embodied AI. This paper contributes to the progress in this area by introducing a causally-inspired hierarchical reinforcement learning (HRL) policy. The use of a newer and larger dataset with complex scene layouts and diverse objects enhances the realism of the task. The results demonstrate the effectiveness of the proposed HRL policy, highlighting the advancements in tackling the challenges of interactive navigation.

+) The decomposition of the embodied policy into a master policy and sub-control policies is a reasonable and effective approach for the IN task. This work demonstrates the potential of HRL in tasks where interactions with the environment are crucial for achieving goals.

+) The explanation of confounding bias and the causal diagram provided in Fig. 2(a) help clarify the concept and its relevance to the IN task. However, the discussion of heavy obstacles as an example may need further refinement.

+) The experimental results presented in the paper provide strong evidence for the effectiveness of CaMP. It outperforms several baselines, including PPO, NIE, and HRL, on the IN task using the newly collected large-scale dataset in ProcTHOR multi-room scenes.

**Weaknesses:**

-) The connection between confounding bias resulting from unmeasurable obstacles and the counterfactual policy design is not adequately explained. It remains unclear how the counterfactual policy, through the intent policy, effectively addresses the bias caused by obstacles. The paper lacks a convincing explanation for the direct line from $O$ to $A$ in Fig. 2(b). It is not clear how the weighted-sum of action logits from the sub-control policies can fully capture the policies' intents and uncover the causality depicted in Fig. 2(a) ($O$ -> $A$ and $O$ -> $R$).

-) The model design lacks intuition. Instead of introducing an additional Intent Policy to generate intents, a more straightforward approach would be to recursively use the Master Policy to obtain intents $P^j(i_t)$ and provide feedback to the Master Policy. This recursive feedback mechanism could be extended to multiple levels, denoted by $j$ ∈ $J$, with $P^0(i_t)$ representing a void intent. Additionally, rather than using the action logits from the sub-control policies to represent intents, exploring the utilization of hidden features returned from the GRU in each sub-policy would offer a more intuitive approach.

-) Several implementation details are missing, such as determining which object to interact with when using Push/Pick actions, the force applied during the Push action, and whether the force is correlated to the object's mass. Further clarification on these aspects would greatly enhance the understanding of the proposed approach. More questions regarding missing details can be found in the Question.

**Questions:**

Please address following points for a more comprehensive understanding of the proposed approach.

Clarifications:

o) Please provide further clarification on the connection between the confounding bias resulting from unmeasurable obstacles and the CaMP model design, specifically the relationship between Sec. 3 and Sec. 4.

o) Clarify why using the weighted-sum of action logits is an effective way to represent intent and its relationship to the confounding bias.

Missing details:

o) How to determine which object to interact with when using the Push/Pick actions?

o) What is the amount of force applied during the Push action, and is it correlated with the object's mass?

o) Provide the exact dimension of the intent embedding $P(i_t)$. Is it equivalent to the size of the total action space?

o) Provide architectural details of the policy, such as whether the CNN is a simple CNN or a CLIP pretrained ResNet, and the number of layers in the GRU.

o) Clarify the meaning of "epochs with a rollout of data" at L191. Does it refer to the number of update iterations using a rollout of data in the PPO implementation in AllenAct?

o) Missing implementation details of the HRL baseline. Does the HRL apply all outputs from the sub-control policies in a single execution, or does it follow a traditional HRL policy where the master policy calls one of the sub-control policies at each step?

Confusions:

o) Clarify whether $r_{tp}$ at L227 is the same as $r_{sp}$ at L229.

o) At L235, it is mentioned that $r_{nav}$ is used to learn the push and pick sub-control policies. This appears to contradict the context provided at L235. Please clarify the requirements for task success, specifically if the agent needs to reach the target position.

**Limitations:**

-) While the proposed CaMP is interesting and evaluated in a complex environment with non-trivial tasks, it would be beneficial to see more results in a simpler environment. For example, evaluating the model on a 2D environment like PettingZoo, where a dot agent performs the interactive navigation task by interacting with obstacles, could provide insights into how the model disentangles intertwined factors by the causally-inspired design in a more controlled setting.

---

> ### Author Rebuttal · Authors · 2023-08-04
>
> We thank the reviewer for the constructive criticism. We address the concerns in detail in the following lines.
>
> Q1. Model design explanation. The reviewer wonders “the connection between confounding bias resulting from unmeasurable obstacles and the counterfactual policy design”, “how the weighted-sum of action logits from the sub-control policies can fully capture the policies' intents and uncover the causality depicted in Fig. 2(a)”, and argues that “a more straightforward approach would be to recursively use the Master Policy to obtain intents and provide feedback to the Master Policy”.
>
> A1. We appreciate the questions and further clarification is detailed as follows.
>
> First, counterfactual decision-making theoretically addresses the problem of learning sub-optimal policy caused by confounding bias from unmeasurable obstacles. Obstacles in InterNav scenario can be regarded as an unobserved confounder (UC) since they influence both the decision-making of actions and the generation of rewards. For instance, the agent may decide to take object interactions when encountering an obstacle (O $\rightarrow$ A). Here, obstacles (O) serve as the mediator from state to action (S $\rightarrow$ A). The causality from action to reward (A $\rightarrow$ R) is confounded by UC, leading to poor estimation of value (accumulated reward) which likely results in sub-optimal policy in RL training. In the theorem of causality inference, unlike observable confounders, UCs can hardly be addressed directly with methods like *Intervention* or *Back-door adjustment*. However, it can be theoretically proved that a counterfactual policy considering intent obtains more value than a standard policy when there exists UC [1].
>
> In Section 4, we apply counterfactual policy to a hierarchical decision framework. In addition to addressing UC, learning counterfactual policy also addresses the indirect feedback problem of master policy by providing it with information about the low-level decision-making through integrated intent.
>
> Second, we believe the sum of actions from sub-control policies weighted by master policy’s decision can fully represent agent’s hierarchical intent, since it contains agent’s intent on each atomic action and the full distributional information of four policies. Since the primary definition of intent is “action before execution” that $I=i_t=f_i(s_t,o_t)$, we don’t see the rationale of implementing intent with hidden features from GRU instead of action logits.
>
> Third, we find the idea of extending the recursive feedback to multiple levels interesting since it may explore the effect of "recursive intent", namely the intent generated based on a priori intent. And we are training new models based on the baseline of PPO to study how a "recursive intent" may help policy learning and will report the results in our final paper (given the limited time for model training). Nonetheless, we believe implementing agent’s intent with an intent policy is reasonable and it’s flexible for us to utilize old intent from iterations behind to balance the policy exploration.
>
> Q2. Missing details and confusions.
>
> A2. Thanks for pointing them out and we will reply to them item by item as follows.
>
> (1) **Q**: "How to determine which object to interact with when using the Push/Pick actions?"
>
> **A**: When taking Push/Pick actions, the object to interact with would be the closest pushable/pickable (predefined according to category) and observable (within 1.25m) object.
>
> (2) **Q**: "What is the amount of force applied during the Push action, and is it correlated with the object's mass?"
>
> **A**: The amount of force applied on the object during the Push action is 100 Newton.
>
> (3) **Q**: "Provide the exact dimension of the intent embedding $P(i_t)$. Is it equivalent to the size of the total action space?"
>
> **A**: The dimension of intent embedding is 12, equivalent to the size of our action space.
>
> 4) **Q**: "Provide architectural details of the policy, such as whether the CNN is a simple CNN or a CLIP pretrained ResNet, and the number of layers in the GRU."
>
> **A**: The CNN is implemented as a simple CNN and the number of GRU layers is 1, which is in line with prior work [2] and the default setting of AllenAct.
>
> 5) **Q**: "Clarify the meaning of "epochs with a rollout of data" at L191. Does it refer to the number of update iterations using a rollout of data in the PPO implementation in AllenAct?"
>
> **A**: “Epochs with a rollout of data” refers to the number of update iterations using a rollout of data in PPO.
>
> 6) **Q**: "Missing implementation details of the HRL baseline. Does the HRL apply all outputs from the sub-control policies in a single execution, or does it follow a traditional HRL policy where the master policy calls one of the sub-control policies at each step?"
>
> **A**: In the HRL baseline, the master policy calls one of the sub-policies (with the same splits of action space of our model) at each step and the action is output by the sub-policy.
>
> 7) **Q**: "Clarify whether $r_{tp}$ at L227 is the same as $r_{sp}$ at L229."
>
> **A**: It’s a typo and both should be step penalty $r_{sp}=0.01$.
>
> 8) **Q**: "At L235, it is mentioned that $r_{nav}$ is used to learn the push and pick sub-control policies. Please clarify the requirements for task success, specifically if the agent needs to reach the target position."
>
> **A**: The reward for interactive auxiliary tasks should be $r_{inter}=r+r_{as}-r{af}$, where $r=r_{success}+\Delta_{dis}-r_{sp}$ and $r_{success}$ is obtained when the goal of interactive task is achieved (taking Done when the obstacle is cleared).
>
> **References**
>
> [1] Zhang, Junzhe, and Elias Bareinboim. Markov decision processes with unobserved confounders: A causal approach. Technical report, Technical Report R-23, Purdue AI Lab, 2016.
>
> [2] Zeng, Kuo-Hao, et al. "Pushing it out of the way: Interactive visual navigation." Proceedings of the IEEE/CVF Conference on Computer Vision and Pattern Recognition. 2021.

---

> ### Author Response · Authors · 2023-08-13
>
> We really appreciate the reviewer for the concerns and criticisms for this work, and hope our response has resolved the confusion and questions in the review. If there are any questions or further comments, please let us know and we will try our best to answer them!

---

> > ### Comment · Reviewer_7DX2 · 2023-08-17
> > **Thank the authors for detailed responses.**
> >
> > I appreciate the authors for their thorough response in addressing my concerns, particularly in clarifying missing details and points of confusion. I do not have further questions regarding to those aspects.
> >
> > I would like to also thank the authors for their detailed responses regarding my questions about "the connection between confounding bias resulting from unmeasurable obstacles and the counterfactual policy design" as well as "uncover the causality depicted in Fig. 2(a)".
> >
> > However, I still find it challenging to be convinced by the statement suggesting that "the sum of actions from sub-control policies weighted by the master policy's decision can fully represent the agent's hierarchical intent." While I understand that action logits can, to a certain extent, capture the agent's intent, I remain concerned that these logits, being the end result of sub-policies, might disregard crucial information about the environment and the agent's comprehension and belief regarding its state. It's possible that this approach might only capture the agent's intentions at isolated time steps and overlook the memory accumulated along the trajectory the agent has traversed.
> >
> > For example, in scenarios involving an obstacle in front of the agent, while the Push-sub-policy aims to push the obstacle, the Navigate-sub-policy might suggests a detour due to its memory of an alternative route. Given that the framework constructs intent solely through weighted-sum of action logits, it may not encapsulate the holistic context such as visited states or environment configuration. Since each sub-policy employs a GRU (as described at L169), it would be insightful to consider a baseline where intent is constructed by summing the weighted GRU memories from sub-policies, rather than just action logits. This could offer valuable insights for future HRL model design for embodied agents.
> >
> > Finally, I want to thank authors for preparing the model with recursive intent. Looking forward to seeing the results! I'm open to raising my score if strong arguments or new findings suggest that the remaining concerns are less impactful than they seem.

---

> > > ### Author Response · Authors · 2023-08-20
> > >
> > > We thank the reviewer for the further concern. Regarding the intuition of model design, one of our purposes of applying the integrated intent is to provide the high-level policy with information of low-level decision-making (L188). We agree that the integrated intent may somewhat overlook the low-level comprehension of agent's state, although the agent can access the state information through the high-level GRU memories. As the reviewer suggested, we modify the CaMP to build a new baseline (denoted as CaMP-mem) where the master policy takes in the sum of memories from sub-policy GRUs weighted by the intent of the master policy:
> > >
> > > $P(i_t)=F(\sum_{\omega^{j} \in \Omega}h_{t}^{\omega^{j}}\cdot P(\omega^{\prime}_{t}=\omega^{j}))$
> > >
> > > where $h_{t}^{\omega^{j}}$  denotes the memory of sub-policy $\omega^{j}$ corresponding to equation (7). We embed the memories with a linear model $F(\cdot)$ to match the size of integrated intent, so that we can train the new model by fine-tuning CaMP with its parameters (given the limited time). We train the CaMP-mem model with 2 million steps on the validation set and report the testing results in the table below (compared with CaMP and its variant without intent):
> > >
> > > |  |  | All |  |  |  | Hard(N$\geq$4) |  |
> > > |:---:|:---:|:---:|:---:|---|:---:|:---:|:---:|
> > > | Methods | SR(%) | SPL | FDT |  | SR(%) | SPL | FDT |
> > > | CaMP wo/intent | 50.8 | 0.294 | 4.03 |  | 31.0 | 0.151 | 6.79 |
> > > | CaMP | 56.3 | 0.327 | 3.67 |  | 41.4 | 0.231 | 5.76 |
> > > | CaMP-mem | 51.2 | 0.298 | 3.96 |  | 32.9 | 0.168 | 6.78 |
> > >
> > > The results illustrate that constructing the intent of action logits obtains better performance than constructing the intent with low-level memories. We speculate the reason is that the knowledge increment from sub-policy memories compared with the master policy memories is not significant enough. However, CaMP-mem is slightly better than CaMP wo/intent, showing the memories from sub-policies are still valuable for high-level decision-making.
> > >
> > > We still thank the reviewer for the useful comments. Moreover, about the comparison in the above table, we think a counterfactual policy should make decisions based on the intent (unexecuted action) rather than the intermediate variables (e.g. hidden features, memories) since the intent is the only "blind spot" of the decision-making process. For instance, it's true that the GRU memories may "encapsulate the holistic context such as visited states or environment configuration". However, even if the agent's memories are not provided through intent, the exactly same memories will be generated by GRU later and help the agent understand the states (regardless of HRL). And the GRU is trained for the purpose of utilizing memories to better encode the states. On the contrary, since the action logits are the end results of the policy, the comprehension of  "last mile" decision-making is invisible for the agent, which makes the intent particularly valuable.

---

> > > > ### Comment · Reviewer_7DX2 · 2023-08-21
> > > > **Thanks the authors for providing the CaMP-mem baseline.**
> > > >
> > > > I appreciate the authors for introducing the new baseline, CaMP-mem, even though it's a quickly finetuned version of the original CaMP model using the validation set. It's surprising to observe that the results from CaMP-mem are worse compared to CaMP, and only marginally better than CaMP w/o intent. Potential explanations for this performance gap could involve (1) the limited number of validation scenes, (2) unnecessary bottleneck creation due to the embedding by $F$, or (3) potential absence of complete model retraining from scratch. Although I'm not entirely convinced by the explanation that the final action logits from sub-policies can sufficiently represent "intent," I acknowledge that the authors made sincere efforts to address my concerns given the time constraints. Apart from this issue, I'm happy with the authors' detailed responses addressing (1) missing details (i.e., model details, training details, environment details), (2) clarifying confusion points (i.e., connection between the confounding bias and the policy design), and (3) presenting the finetuned CaMP-mem baseline.
> > > >
> > > > Overall, my primary concern about the policy design still remains to some extent. I have adjusted my score to Borderline Accept to acknowledge the authors' rebuttal efforts.

---

> > > > > ### Author Response · Authors · 2023-08-21
> > > > >
> > > > > Thanks for the acknowledgment and useful suggestions. We'll revise the final version according to the suggestions and our responses.

---

### Official Review · Reviewer_HVkt · 2023-07-11

**Soundness:** 3 good
**Presentation:** 3 good
**Contribution:** 3 good
**Rating:** 7
**Confidence:** 4

**Summary:**

This paper tackles the problem of interactive visual navigation; i.e., an agent navigating in an enviornment where it is allowed to affect the configuration of the environment (e.g., by moving objects around or picking them up), to improve navigation performance. The key idea is to learn a hierarchical policy that factors in agent intent to propose an action that either results in navigation towards the goal, or interaction with an obstacle. A dataset based on the PROC-THOR simulation environment is also introduced, to facilitate evaluation.

**Strengths:**

[S1] The proposed method is sound; and for the most part, well-defined. The problem formulation is easy to follow, and the description of the method is clear.

[S2] Interactive navigation is a challenging and relevant robotics/embodied-AI problem to the Neurips community. While a large fraction of existing approaches focues on non-interactive visual navigation, this paper explicitly considers affecting state changes by manipulating obstacles, resulting in a novel problem setting where there isn't much prior work.

[S3] I find the positioning of this paper w.r.t. existing literature fair. The baseline methods considered for evaluation are representative of the various flavors of non-interactive visual navigation approaches that have, over the years, been proposed for PointGoal navigation.

[S4] The paper discusses enough implementation details that a reasonable practitioner may be able to replicate the key aspects of the model architecture and the state, action, reward structures.


**Weaknesses:**

Meta-comment: I have one major concern with the experiment design and evaluation setup; which unfortunately results in the key claims of the paper not being substantiated. I have tried to elaborate the issue, and also the rationale behind it. I also suggest a few mitigation strategies (note: these aren't the only possible ones; other strategies welcome too). While I am unsure whether these may be addressed in the short author response window, if these are adequately addressed, I would have no reservations in bumping my score up.


[W1] **[Major] Evaluation metrics and experiment design**:

Interactive navigation is very tricky to evaluate. In (non-interactive) navigation scenarios, the path length metrics used in literature are often highly correlated to execution time. Assuming that each atomic action executed by the agent takes nearly the same time to execute, a longer path length would mean that the agent takes longer to reach the goal. However, in the interactive navigation scenario, typical path length metrics like SPL are no longer good indicators of the amount of time it would take to complete the task. This is because, the time spent in picking up or moving objects will also count towards the overall "time-to-reach-goal-state"; and picking up or moving objects is heavily dependent on object states, which voids the assumption that each action takes roughly the same amout of time.

I understand the rationale behind using typical metrics like SR (success rate), SPL (success-weighted inverse path length), and FDT (final distance to target) -- these metrics allow for easy benchmarking with existing approaches (esp. non-interactive navigation techniques); these are also easily accessible via modern simulators that support PointGoal navigation.

That said, reporting only these above metrics only portrays the benefits of an interactive navigation strategy, while masking away the disadvantages. Execution time is often lost when interacting with objects, which goes unaccounted for. This becomes apparent when you consider the following counterexample: assume that an agent spends more than half its time moving (or picking up) every object encountered in its way; but ends up taking the shortest possible (i.e., optimal) path to goal. It will then end up with a success rate and SPL of 1 and an FDT value of 0. However, in reality, an agent that takes a twice as long path to the goal (but does not interact with any object) will end up reaching the goal at about the same time, and will have the same SPL, SR, and FDT values (recall that, in this hypothetical scenario, the interactive agent spends half its time interacting with obstacles in its path). This calls for a different approach to evaluate interactive navigation approaches; ideally the SPL metric should also account for time lost due to interaction. Measuring this time could be challenging (or impossible) to do in the first place, so here are a few potential mitigation strategies to consider.

* While episode lengths are fixed, to ensure the agent does not take an indefinite amount of time, to compare fairly against non-interactive agents, each episode length must be capped to the number of timesteps in which a non-interactive agent completes the task. This will ensure that a fair comparison is possible, when a reasonably accurate time estimate is available for each interaction.
* An alternative would be to consider evaluating solely in scenarios where it is impossible to reach the goal without interacting with objects. (more nuance on this follows in [W2] below).

[W2] **Eval on data subsets**: I would have liked to seen more granularity in terms of the quantitative results presented in Table 3. It would, for instance, be useful to split the dataset into a variety of categories, depending on whether or not interaction is required to solve an episode, and the level of difficulty of the episode. This becomes more important, because the dataset is claimed as a contribution. Table 2 goes a bit along this direction, indicating the number of obstacles per room; however, it is not clear how many of these episodes absolutely require interactive navigation (i.e., cannot be solved by a non-interactive optimal agent). (The dataset construction seems to ensure a 50% likelihood that a path is unavailable -- I would argue in favor of forming two splits of the dataset, one where no non-interactive agent trajectory exists; and one where an optimal non-interactive agent will need to take a longer path).


[W3] **[Minor]** The paper, in its current form, falls short in terms of technical rigor when discussing aspects of causal inference. E.g., lines 134-135 "However, without considering the causalities from obstacles, it’s hard to generalize to unseen environments or large-scale datasets."; line 38 "... , learning causality through RL training is challenging due to the existence of unobserved..."; line 137 "The above confounding bias can be tackled...". Neither the "causalities stemming from obstacles" nor the "confounding bias" have been clearly been defined in the paper. It is also important to clearly define the type of causal relationships that are being learned by design (perhaps section 4.2 is a good place to do so).
Revising these aspects of the paper will make the descriptions of the approach more accurate.

**Questions:**

Overall, I like where this paper is going. The hierarchical policy presented herein is sound. My concerns are not with the method; rather with the choice of evaluation metrics. In my review (see weaknesses above), I have attempted to also provide a few mitigation strategies (these are by no means extensive -- merely initial thoughts on how newer evaluation metrics may be designed for the task at hand).

I would like to see this discussed further in the author response phase. If addressed adequately, I have no hesitation in bumping my score up.

**Limitations:**

Adequately addressed

---

> ### Author Rebuttal · Authors · 2023-08-03
>
> We thank the reviewer for the constructive concerns and address them in detail in the following lines.
>
> Q1. Evaluation metrics and experiment design.
>
> The reviewer is concerned about the evaluation of InterNav that “in the interactive navigation scenario, typical path length metrics like SPL are no longer good indicators of the amount of time it would take to complete the task.” and “That said, reporting only these above metrics only portrays the benefits of an interactive navigation strategy, while masking away the disadvantages.” and provides a few potential mitigation strategies of evaluation.
>
> A1. We agree that “time of completing the task” is also an important evaluation indicator for InterNav. In our scenario, by calculating the standard time cost of each action (including movement and manipulation), a time cost metric can also be evaluated in our experiments, which can be regarded as a time-measurement variant of the SPL metric. Also, considering that most previous InterNav works [1,2] report the SPL, in order to compare with those works directly, we’ll report both time and path length metrics in the following response and final version.
>
> First, we evaluate an additional metric STS (short for Success rate weighted by Time Steps) to measure the time cost of task completion: $STS=\frac{1}{N}\sum_{n=1}^{N}Suc_{n}\frac{L_{n}/grid}{TS_{n}}$, where $L_{n}$ is the shortest path length, $TS_{n}$ is the timesteps agent takes to complete the task, and $grid=0.25m$ is the unit distance of agent moving forward in one step. Thus $L_{n}/grid$ represents the number of timesteps it takes to navigate to the goal by merely moving forward (without spawned obstacles). Since the agent takes atomic actions in AI2-THOR simulator and each of them shares the same amount of time to execute (a timestep), we measure the time cost with the number of timesteps. STS is higher when the agent accomplishes the task with less time and the ideal situation is that the goal is directly ahead and there is no need for interaction where STS is 1 (it's the most ideal situation for all navigation scenarios so that it's computable for InterNav). As a matter of fact, we train our model following that idea, since our reward shaping $r=r_{success}+\Delta_{dis}-r_{tp}$ encourages interaction that efficiently reduces the goal distance with fewer timesteps, rather than pursuing a shorter trajectory. Thus, both strategies of efficient bypass and effective interaction are rewarded. We report the performance of several models in the table below, and the result shows that STS is a more stringent measure and is able to reflex the performance difference between models.
>
> |Methods|STS (all)|STS (N$\geq$4)|
> |:----:|:----:|:----:|
> |PPO|0.134|0.086|
> |NIE|0.155|0.102|
> |HER|0.135|0.084|
> |PPO+intent|0.163|0.109|
> |CaMP|**0.177**|**0.121**|
>
> Second, in InterNav scenarios, SPL is an indicator of agents' ability of object interaction, which is crucial especially in cluttered environments. Fei Xia et al. [2] have introduced SPL to interactive navigation as the measurement of path efficiency that “Path Efficiency: how efficient the path taken by the agent is to achieve its goal. The most efficient path is the shortest path assuming no interactable obstacles are in the way.” Although the most efficient path may not correspond to the least time cost, it indicates the most effective interaction. For InterNav in complex multi-room environments, we aim at improving agent's interactive ability so it can proactively change the environment for better navigation, rather than limiting its strategic choices to its capacity.
>
> Q2. Eval on data subsets. The reviewer suggests that "I would argue in favor of forming two splits of the dataset, one where no non-interactive agent trajectory exists; and one where an optimal non-interactive agent will need to take a longer path".
>
> A2. We appreciate the suggestion and form a split of dataset where non-interactive trajectories (longer than the shortest path) exist to enrich the evaluation. We first calculate the ratio of that split in the whole dataset: 20.5% (overall), 27.4% (1 $\sim$ 2 rooms), 18.6% (3 $\sim$ 5 rooms), 12.4% (6 $\sim$ 10 rooms). Then we report the performance of models (a non-interactive PPO trained on ProcTHOR is included) on the **new split** in the table below. It's interesting to find that PPO without interaction achieves better STS (0.201) compared with PPO (0.181), although it obtains lower SR on the non-interactive set (47.4%) and the whole set (21.5%). It indicates that the strategy of object interaction may cost unnecessary time in uncrowded environments and the agent needs to balance the efficiency and efficacy during the task.
>
> |Methods|SR (%)|SPL|STS|FDT|
> |:----:|:----:|:----:|:----:|:----:|
> |PPO (non-inter)|47.4|0.309|0.201|4.62|
> |PPO|51.5|0.306|0.181|3.44|
> |NIE|58.8|0.345|0.188|3.01|
> |HER|51.9|0.316|0.176|3.40|
> |PPO+intent|70.4|0.390|0.222|2.38|
> |CaMP|**72.3**|**0.407**|**0.236**|**2.05**|
>
> Q3.  Concept clarification.
>
> A3. Thanks for the concern. By "causalities from obstacles" we refer to the causal relationships from the obstacle (O) to other causal factors (i.e. O $\rightarrow$ A, O $\rightarrow$ R in Figure 2). By "confounding bias" we refer to the negative phenomenon of $P(R|do(A))\neq P(R|A)$ caused by the confounder (i.e. obstacles in InterNav). And our method is designed to better learn the causality from action to reward (A $\rightarrow$ R). We will revise the concept clarification of our paper for better understanding.
>
> **References**
>
> [1] Zeng, Kuo-Hao, et al. "Pushing it out of the way: Interactive visual navigation." Proceedings of the IEEE/CVF Conference on Computer Vision and Pattern Recognition. 2021.
>
> [2] Xia, Fei, et al. "Interactive gibson benchmark: A benchmark for interactive navigation in cluttered environments." IEEE Robotics and Automation Letters 5.2 (2020): 713-720.

---

> ### Author Response · Authors · 2023-08-11
>
> Thanks to the reviewer for the concerns and suggestions for this work. We hope our response has resolved the confusion and questions in the review. If there are any questions or further comments, please let us know and we will try our best to answer them!

---

> > ### Comment · Reviewer_HVkt · 2023-08-13
> > **Author response effectively addresses all major concerns**
> >
> > Hi authors,
> >
> > Many thanks for all your (substantial) efforts in preparing the author response. This addresses all of the major concerns I had on the initial manuscript.
> >
> > I have read through the other reviews and responses therein. My concerns were not so much about the soundness of the work (as I stated in my initial review, the method was sound); it was only with the design of the experiments and evaluation metrics. The STS metrics and the analyses on the new split address both these concerns. As such, I have no hesitation in further improving my score.
> >
> > The somewhat contrary trends across STS and SR on the new split indicate that there is no one metric yet that may be used as a standalone indicator of performance, and that we may need both STS and SR to be high, when determining which InterNav approach is better. This is worth clarifying in a revised manuscript.
> >
> > Again, I really appreciate this response; and like where this work is heading.

---

> > > ### Author Response · Authors · 2023-08-21
> > >
> > > Thanks for the appreciation and useful suggestions. We'll revise the final version according to the suggestions and our responses.

---

### Decision · Program_Chairs · 2023-09-21

**Decision:**

Accept (poster)

**Comment:**

This work introduces a new dataset of multi-room interactive navigation tasks, along with a causal, hierarchical, multi-policy framework for addressing these tasks. The framework consists of a high-level policy, an intent policy, and sub-control policies. The authors conduct extensive benchmarking and analysis of their framework on the ProcTHOR platform.

The reviewers highlight several strengths of this work, including:
* The dataset is timely, challenging, and interactive, and provides a standardized platform for the embodied AI community.
* The causal, hierarchical, multi-policy framework is empirically effective, as demonstrated by the extensive benchmarking.
* The motivation and presentation of the approach are clear and instructive.

The AC concurs with the unanimous recommendation of accepting this submission.

The authors have constructively discussed several writing and technical clarifications, new results, new data splits, etc. with the reviewers. These changes would improve the value and impact of the paper. A minor edit, that was missed in the reviews, could be a more inclusive re-naming of the current high-level ("master") policy.